# Attention-Enabled Ensemble Deep Learning Models and Their Validation for Depression Detection: A Domain Adoption Paradigm

**DOI:** 10.3390/diagnostics13122092

**Published:** 2023-06-16

**Authors:** Jaskaran Singh, Narpinder Singh, Mostafa M. Fouda, Luca Saba, Jasjit S. Suri

**Affiliations:** 1Department of Computer Science, Graphic Era, Deemed to be University, Dehradun 248002, India; jaskaran.jsk2001@gmail.com; 2Department of Food Science and Technology, Graphic Era, Deemed to be University, Dehradun 248002, India; narpinders@yahoo.com; 3Department of Electrical and Computer Engineering, Idaho State University, Pocatello, ID 83209, USA; mfouda@ieee.org; 4Department of Neurology, University of Cagliari, 09124 Cagliari, Italy; lucasabamd@gmail.com; 5Stroke Monitoring and Diagnostic Division, AtheroPoint™, Roseville, CA 94203, USA

**Keywords:** depression, ensemble deep learning, attention-enabled, diagnosis, domain adoption

## Abstract

Depression is increasingly prevalent, leading to higher suicide risk. Depression detection and sentimental analysis of text inputs in *cross-domain frameworks* are challenging. Solo deep learning (SDL) and ensemble deep learning (EDL) models are not robust enough. Recently, *attention mechanisms* have been introduced in SDL. We hypothesize that *attention-enabled* EDL (*ae*EDL) architectures are superior compared to *attention-not-enabled* SDL (*ane*SDL) or *ae*SDL models. We designed EDL-based architectures with *attention blocks* to build eleven kinds of SDL model and five kinds of EDL model on four domain-specific datasets. We scientifically validated our models by comparing “seen” and “unseen” paradigms (SUP). We benchmarked our results against the SemEval (2016) sentimental dataset and established reliability tests. The mean increase in accuracy for EDL over their corresponding SDL components was 4.49%. Regarding the effect of attention block, the increase in the mean accuracy (AUC) of *ae*SDL over *ane*SDL was 2.58% (1.73%), and the increase in the mean accuracy (AUC) of *ae*EDL over *ane*EDL was 2.76% (2.80%). When comparing EDL vs. SDL for non-attention and attention, the mean *ane*EDL was greater than *ane*SDL by 4.82% (3.71%), and the mean *ae*EDL was greater than *ae*SDL by 5.06% (4.81%). For the benchmarking dataset (SemEval), the best-performing *ae*EDL model (ALBERT+BERT-BiLSTM) was superior to the best *ae*SDL (BERT-BiLSTM) model by 3.86%. Our scientific validation and robust design showed a difference of only 2.7% in SUP, thereby meeting the regulatory constraints. We validated all our hypotheses and further demonstrated that *ae*EDL is a very effective and generalized method for detecting symptoms of depression in cross-domain settings.

## 1. Introduction

Depression is a serious and debilitating mental health condition that affects millions of people worldwide, affecting 260 million people globally [1]. According to the National Institute of Mental Health, depression is increasingly prevalent, affecting individuals’ ability to function in daily life, resulting in suicide risk increasing by 35.2% from 2000 to 2020 [2]. It is characterized by persistent feelings of sadness, hopelessness, and a loss of interest in daily activities [3]. Individuals with depression often experience a range of physical and emotional symptoms, including fatigue, insomnia, changes in appetite, and difficulty concentrating in day-to-day activities [4,5]. Depression can have a significant impact on an individual’s quality of life, affecting their personal and professional relationships, their ability to work, and their overall sense of well-being [6,7]. Therefore, early detection is essential to prevent the condition from worsening and to help individuals access appropriate treatment and support. Through this, individuals may be better able to manage their symptoms and maintain their ability to work and contribute to society. This can lead to better outcomes for both individuals and society as a whole [8,9].

Depression detection has been conducted for over 200 years through the identification of an individual’s behavior by qualified psychologists [10,11]. Machine learning (ML) has become very popular in healthcare, particularly in the field of the classification of diseased vs. control patients [12,13]. Several studies have explored the use of statistical ML models to categorize a person’s chats and texts as exhibiting either depressive or non-depressive behavior by analyzing patterns in language use and to identify features that are indicative of depression [14,15]. As observed before, ML models suffer from poor performance due to their inability to handle the non-linearity of risk predictors and gold standard labels or events [13,16,17]. Similarly, the linear structure architecture of current automated depression detection models renders them susceptible to poor performance, since they only focus on individual words and fail to consider the context of previous and subsequent words. These models tend to be slow, due to non-parallel and slow processing, and offer few options for algorithmic tuning and refinement.

Deep learning (DL) has rapidly gained momentum in large number of applications due to its automated ability to extract automated features [18]. These models utilize fully connected layers with neurons and activation functions, creating networks that mimic the human brain’s functioning [19]. Recently, advanced DL models have penetrated the field of text classification and are capable of identifying complex sequences in language data [20,21,22]. The use of DL models and open-source embedding techniques, such as Word2vec and GloVe [23], have shown promise in addressing the challenge of detecting depression. By using embeddings, text data can be converted into dense vectors, where semantically similar inputs are located close to each other [24]. The introduction of architecture such as Gated Recurrent Units (GRUs) has improved the results of depression detection [25], but there are still limitations, such as the nature of a single input–output channel and the inability to achieve optimal results through a single base classifier. 

Ensemble deep learning (EDL) represents a breakthrough in the field of DL, providing the potential for better performance than standalone models [26,27]. It enables the training of data of varying sizes, shapes, and types to different base classifiers and produces a single predictive output, which may be helpful in situations where data are of a multimodal nature [28,29,30]. Studies, including [31], have employed clustering and ensemble-based models to yield superior results in sentiment detection. To further enhance the performance of EDL, incorporating attention channels (or blocks) into the model architecture could increase its robustness and enable a more focused analysis of specific input tokens [32,33,34]. By identifying key features within the input data, the *attention mechanism* could potentially improve the accuracy, reliability, and generalizability of the model, particularly in applications related to mental health or other complex domains. Study [35] employed an attention-enabled LSTM model for sentiment analysis at the document level, which included a joint loss function to enhance its performance. Additionally, transformers have been widely used for sentiment analysis, as demonstrated by another study [36] that utilized a weight ensemble of transformer models to detect aggressive text in the Bengali language, employing various BERT-based techniques. 

Multi-head co-attention networks enable us to attend independently to different parts of a sequence. Study [37] leveraged this approach to perform aspect-level sentiment analysis on a text dataset, surpassing existing methods. In the same domain of aspect-based sentiment extraction, numerous studies have explored triplet extraction techniques involving target, opinion, and sentiment extraction [38,39].

To conduct our study, we constructed eleven attention-enabled solo deep learning (*ae*SDL) and five attention-enabled ensemble deep learning (*ae*EDL) models and evaluated their performance on four main datasets, namely, two public datasets (SD-Sford-09 and DD-Kgg-22), and two proprietary datasets (DD-Red-14 and SD-Twi-2). We utilized self-attention blocks to determine the overall performance improvement after incorporating attention mechanisms into the models. Additionally, we calculated the average performance gain of *ae*EDL versus *ae*SDL, as well as *attention models* versus *non-attention,* or without attention (wa), models. We conducted a benchmark of our model on two public datasets and achieved the best performance with our *ae*EDL (ALBERT+BERT-BiLSTM), outperforming previous studies in the literature. Furthermore, we validated our model using various statistical tests and cross-validation protocols on seen vs. unseen datasets to verify the robustness of our *ae*EDL. Finally, we conducted cross-domain tests to demonstrate the adaptability of our model by training and testing it on datasets with differing semantics.

This paper follows a systematic flow, beginning with Section 2, which describes the methodology. Section 2.1 includes a discussion of the four types of dataset used in the study, the architecture and building of both the SDL and EDL models are outlined in Section 2.2, and the experimental protocols undertaken are discussed in Section 2.3. Next, the paper describes the performance metrics used throughout the study in Section 2.4. The Results section of the paper reports the findings of the study, which are presented across five different experimental protocols in Section 3. Section 4 presents a performance evaluation, whereby ROC curves and bar graphs showcase the models’ performance in Section 4.1, and a discussion of the statistical tests used is presented in Section 4.2. Finally, the paper discusses how the study’s findings compare with other related research studies in Section 5. It also includes a discussion of the study’s principal findings, and a brief note on attention, the strengths and weaknesses, and possible extensions of our current study.

## 2. Methodology

Our methodology involved using simple DL models as a starting point, since they have proven to be effective in various natural language processing (NLP) tasks. To evaluate a model’s efficiency and generalizability, we conducted tests on multiple intra- and cross-domains. Therefore, the first step in our methodology was to collect multiple datasets. Next, we constructed the architecture of the individual models, and then, used them to build the *ae*EDL. Finally, we declare the experiment protocols that we implemented, as well as the performance metrics that we utilized to evaluate the models.

### 2.1. Data Types and Their Preparation

The methodology employed in this study involved gathering data from multiple sources and domains. To conduct our experiment, we collected data from four different sources, two of which were publicly available, and two of which were proprietary. Finally, the fifth sentimental dataset, the famous SemEval (2016) [40], was used for benchmarking.

#### 2.1.1. Dataset 1: SD-Sford-09 

This dataset, “Sentiment140”, labeled “SD-Sford-09”, comprises sentimental data (SD) from Stanford University (Sford), first published in 2009. This is a publicly available dataset [41], which contains 1.6 million tweets, each labeled with the polarity of the tweet, as portrayed in Table 1. A polarity value of zero indicates a negative tweet, while a value of four indicates a positive tweet. The dataset is well balanced, with 800,000 members in each class, and our analysis focused solely on the polarity and text content of each tweet.

#### 2.1.2. Dataset 2: DD-Red-14

We adopted the methodology followed by study [42] to create a depression-centric dataset using the PushShift API to download information from 12 subreddits focused on mental health (such as r/bipolarreddit, r/socialanxiety, r/healthanxiety, r/ptsd, r/autism, r/schizophrenia, r/addiction, r/adhd, r/anxiety, r/alcoholism, r/lonely, and r/depression) and 11 subreddits focused on non-mental health-related topics (such as r/jokes, r/gaming, r/india, r/music, r/teaching, r/legaladvice, r/mildlyinteresting, r/unexpected, r/space, r/cats, and r/news). Subreddits are individual communities within the larger Reddit platform, where users can join and participate in discussions centered around specific topics. These communities often have their own rules, moderators, and user base, creating a unique environment for sharing and interacting with content. To address the validation of the ground truth label, we specifically obtained the dataset from specialized mental health subreddits. These subreddits were externally moderated by dedicated subreddit moderators, who played a crucial role in ensuring the data’s quality. It was labeled “DD-Red-14” since it comprised depression data (DD) from a Reddit source (Red) and was first published in 2014. A total of 13,000 posts were collected from the mental health subreddits and labeled ‘depressive’, and an additional 13,000 posts were collected from the non-mental health-related subreddits and labeled ‘neutral’, as portrayed in Table 2.

#### 2.1.3. Dataset 3: DD-Kgg-22

This dataset was extracted from the Kaggle platform and contains 27,977 posts which comprise text related to people suffering from anxiety, depression, and other mental health issues [43]. It was labeled “DD-Kgg-22”, since it comprises depression data (DD) from the Kaggle (Kgg) source and first published in 2022. Of these, 14,139 entries are from people free from any mental health issues, labeled 0, while 13,838 entries are from people who are suffering from mental health issues, labeled 1, as visualized in Table 3. 

#### 2.1.4. Dataset 4: SD-Twi-23 

This dataset contains 31,000 tweets published between January 2018 and January 2021. It was labeled “SD-Twi-23” since it comprises sentimental data (SD) taken from Twitter (Twi) and first published in 2023. Of these, 16,000 tweets were labeled “negative” and were extracted using keywords such as ‘sad’, ‘bad’, and ‘negative’. An additional 15,000 tweets labeled “neutral” were collected without any filters to serve as a control group for the analysis.

#### 2.1.5. Dataset 5: Benchmarking Dataset—SD-SemEval-16

This dataset contains tweets with the tags “Positive” and “Negative” from SemEval-2016 Task 4 Subtask B [40]. It was labeled “SD-SemEval-16” since it comprises sentimental data (SD) taken from the 2016 competition. The tweets were categorized into the categories train, dev, devtest, and test, with 14,042 labeled as having positive sentiment and 3677 as having negative sentiment, as visualized in Table 4. 

Figure 1 depicts the overall architecture of our study. We began by collecting datasets using published resources and APIs such as Twint. The first block was the pre-processing of the dataset prior to the application of the DL models. In this research paper, we describe the data preprocessing steps used to prepare raw text data for machine learning tasks in natural language processing. Firstly, we performed data cleaning by converting the input to lowercase and removing punctuation and symbols. Then, we tokenized the input paragraphs using word tokenization to convert the sentences into a stream of tokens that can be passed to the machine. Lemmatization was performed to convert words to their base form, or lemma, while retaining their inherent meaning. Stop words were removed from the tokens, including articles, pronouns, and conjunctions. Finally, we performed embedding to map the processed input to its vector counterpart. Embedding is necessary to represent text data as vectors in a high-dimensional space, and we used Word2vec and pre-trained BERT embedding techniques to create a distributed representation of words that capture semantics and relationships among the words.

The power of AI was used once the data preparation had been conducted. Here, we divided the dataset into training and testing sets, and then, built the training models for the (a) SDL models and (b) EDL models. These training models were then used to transform the test datasets, yielding the prediction labels, which were then used for performance evaluation and explainability using the explainable AI module.

**Figure 1 diagnostics-13-02092-f001:**
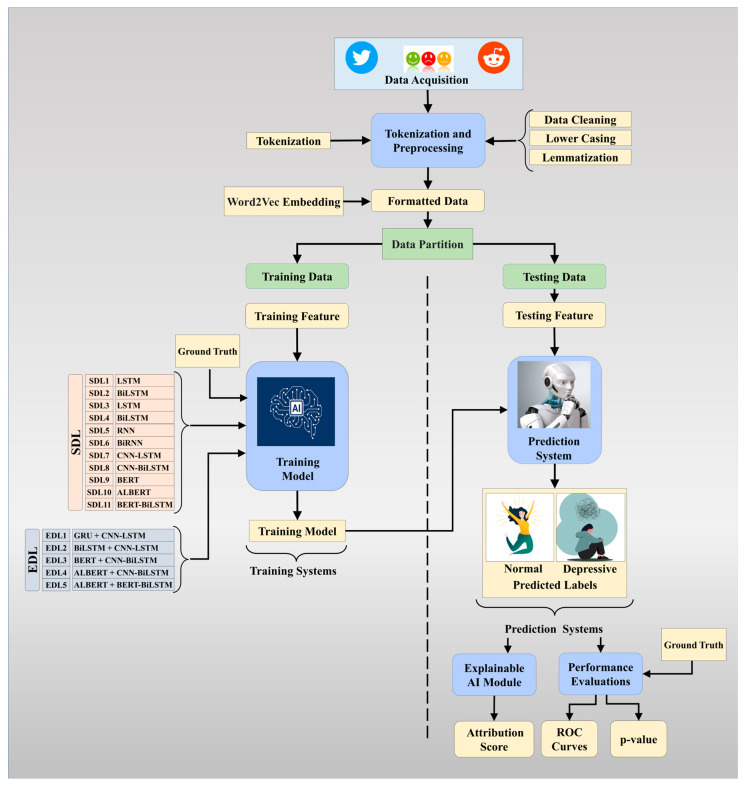
Overall architecture of our proposed study.

### 2.2. Solo Deep Learning and Ensemble Deep Learning Architectures

For our preliminary data analysis, we built a total of 16 models, which included 11 SDL and 5 EDL models constructed from the SDL models. Among the SDL models, we developed three unidirectional models: Long Short-Term Memory (LSTM), Gated Recurrent Unit (GRU), and recurrent neural network (RNN), along with their corresponding bidirectional versions: Bidirectional LSTM (BiLSTM), Bidirectional GRU (BiGRU), and Bidirectional RNN (BiRNN), comprising six models. Additionally, we created two pre-trained SDL models: Bidirectional Encoder Representations from Transformers (BERT) [44] model and its optimized version: A Lite BERT (ALBERT) [45] from Huggingface. BERT and ALBERT were chosen over other models, such as DistilBERT [46] and XLNet, [47] for several reasons. While DistilBERT and XLNet offer comparable performance with faster inference times [48], the larger model sizes and training procedures of BERT and ALBERT provide more powerful representations, capturing complex contextual information and improving overall performance. This advantage was particularly important for our cross-domain approach across all four datasets, as BERT and ALBERT exhibited better adaptability and understanding of text nuances across different domains [49]. 

In addition to their advantages mentioned earlier, BERT and ALBERT excel in their ease of incorporating attention mechanisms and additional attention channels. Their complex architectures include multiple attention heads that can attend to different parts of the input, enabling straightforward extension of the models with extra attention layers and refinement of the attention mechanisms. Furthermore, BERT and ALBERT offer a wide range of pretrained models, providing flexibility and adaptability to meet various experimental requirements. We were able to choose from various model sizes and variations, allowing for customization and optimization based on our specific needs.

While individual deep learning models have shown limited success in detecting depression, combining them into hybrid deep learning models has been shown to improve performance and overcome data scarcity [50]. By leveraging multiple architectures, Hybrid models can address domain-specific challenges and improve accuracy in tasks such as detecting depression [51,52,53,54]. Using these spirits of HDL, we finally constructed three hybrid deep learning (HDL) models: a CNN-LSTM, a CNN-BiLSTM, and a BERT-BiLSTM. These constituents have a total of eleven SDL models.

We constructed five EDL models using fusion through and concatenation with seven SDLs. These EDL models, designed to surpass their individual SDLs, are drawn in Figure 2, Figure 3, Figure 4, Figure 5 and Figure 6, and their constituents are detailed in Table 5.

**Table 5 diagnostics-13-02092-t005:** Constructed EDL models and their components.

SN	EDL Types	EDL Models
1	EDL1	GRU + CNN-LSTM
2	EDL2	BiLSTM + CNN-LSTM
3	EDL3	BERT + CNN-BiLSTM
4	EDL4	ALBERT + CNN-BiLSTM
5	EDL5	ALBERT + BERT-BiLSTM

While all of these architectures follow a similar skeleton, it is the combination of different SDL models, the sequence of layers in each SDL model, and their embeddings that differentiate the architecture and affect its training. To ensure that the anticipated improvement gained is a general trait and independent of the constituent models’ architecture, it was necessary to create five different EDL models. We anticipate that the performance of the EDL models will vary due to their core being built out of different SDL models. Hence, the EDL models will retain some behavior of their original SDL model, which would be optimized in the EDL architecture.

All of the architectures have two modules that can accept two identical or different sets of input tokens. The modules converge into a concatenation layer, followed by a depression detection module that consists of max pooling, dropout, and a fully connected dense layer network. A final output sigmoid layer is used to classify the input as depressive/non-depressive or positive/negative sentiment, depending on the protocol being carried out.

The EDL1 model utilizes two modules: GRU and CNN-LSTM; these are visualized in Figure 2. The GRU layer contains a self-attention layer followed by dense and dropout layers. The CNN-LSTM module includes a convolution layer and a max pooling layer, which is connected to an LSTM layer. The EDL2 model employs the same CNN-LSTM module, along with a BiLSTM module that has self-attention enabled within it, as shown in Figure 3. EDL3 and EDL4 use a CNN-LSTM module with convolution layers along with a BiLSTM module, but with Word2vec embedding being replaced by pre-trained BERT and ALBERT embedding, respectively, as visualized in Figure 4 and Figure 5. EDL5 incorporates both ALBERT and BERT-BiLSTM modules, where each module includes a self-attention layer. These two modules are concatenated to form the depression detection module. Additionally, the BERT module also contains a BiLSTM layer, as depicted in Figure 6.

Attention mechanisms in deep learning are used to improve model performance by allowing the model to selectively focus on important parts of the input. By doing so, attention mechanisms can improve the accuracy of the model’s predictions and reduce training time. Additionally, attention mechanisms can increase the interpretability of the model by providing insight into which parts of the input are most relevant to a given prediction. Hence, we applied attention layers using multi-head self-attention to all the SDL and EDL models to observe the effects of incorporating attention into the models.

Multi-head self-attention is a variant of the self-attention mechanism that involves computing multiple attention heads in parallel and concatenating the output of each head before applying a linear transformation. The multi-head self-attention mechanism that we incorporated can be mathematically illustrated as: 

#### Attention Score Computation

Let X = [x_1_, x_2_, …, x_n_] be the input sequence of length n, and let H = [h_1_, h_2_, …, h_n_] be the output sequence of the multi-head self-attention layer, where h_i_ is the representation of the ith element in H.

First, we compute Query (Q), Key (K), and Value Vectors (V), which are learnable parameters, and d_ks_ is the dimensionality of the query. Then, we calculate the attention scores using the SoftMax function *s* and multiply it with V to obtain the weighted sum for each head:(1)$$H_{i}=s\;(\frac{{\mathrm{QK}}^{T}}{d_{\mathrm{ks}}})\;V_{i}$$

Then, we compute the output sequence using the concatenation of each head, where © denotes the concatenation function: 
H = © ([H_1_, H_2_, H_3_, ….., H_n_])
(2)


Lastly, we apply a linear transformation to map the output and obtain the desired output size (Final Output sequence) H_os_ from H, where it is the desirable size for the next layer: H_os_ = H × *a*(3)
where *a* is a learnable weight matrix for mapping the output of the layer to the desired output size.

### 2.3. Training and Loss Functions

The models in the study were trained using a batch size of 128 and an input layer of 100 tokens, with a binary cross-entropy loss (CLE) function. Binary CLE is a loss function used in ML for binary classification problems. Cross-entropy is a mathematical function that is defined in terms of the logarithm of the predicted label and the gold standard label. It measures the difference between the predicted probabilities of the positive class and the true labels, and penalizes the model for large errors. The binary cross-entropy loss function is denoted as L_bce_, and mathematically, it can be expressed as:(4)$$L_{\mathrm{bce}}=-\frac{1}{N}\times \sum_{i=1}^{N}(Y_{i}\times \log({\hat{Y}}_{i})+(1-Y_{i})\times \log(1-{\hat{Y}}_{i}))$$
where N is the number of samples in the dataset, Y_i_ is the ground truth label (either 0 or 1), ${\hat{Y}}_{i}$ is the predicted probability of the positive class, log is the natural logarithm, and × means multiplication. 

Table 6 provides information on the epochs each model took, as well as their initial learning rates and optimizers. As shown, EDL1 and EDL2 used Adam optimizers, while EDL3, EDL4, and EDL5 used SGD optimizers. The models were trained and tested on a 9:1 split using the K10 protocol. The initial learning rate for EDL1, EDL2, and EDL3 was 2 × 10^5^ and for EDL4 and EDL 5, it was 1 × 10^4^. Finally, it should be noted that EDL 1 was trained for 30 epochs, EDL 2 and EDL 3 were trained for 40 epochs, EDL 4 was trained for 45 epochs, and EDL 5 was trained for 50 epochs. The study was implemented using Python 3.8 and a TensorFlow framework. To implement the system, a 12 GB NVIDIA P100 16 Graphics Processing Unit (GPU) was utilized. Additionally, the system was equipped with an Intel Xeon Processors processor and 12 GB of RAM.

### 2.4. Experimental Protocols

Based on our preliminary analysis and introduction, we developed an experimental workflow, which is outlined in this section. Initially, we examined the SDL models and compared the advantages of bidirectional models over unidirectional models. Subsequently, we investigated how combining SDL models with EDL models can enhance performance on standard datasets. We then evaluated the impact of adding an attention layer to these models on the overall performance of the depression and sentiment analysis task. Finally, we cross-validated our observed results and demonstrated the domain adaptability of our system by performing an unseen paradigm (a situation where the deep learning model is tested on a new and previously unseen task or dataset that is significantly different from the data it was trained upon).

#### 2.4.1. Experiment 1: Unidirectional vs. Bidirectional SDL Models

We conducted this experiment on the SDL model, comparing the performance metrics of unidirectional models versus their bidirectional counterparts (LSTM vs. BiLSTM, GRU vs. BiGRU, and RNN vs. BiRNN) averaged across our four main datasets (SD-Sford-09, DD-Red-14, DD-Kgg-22, and SD-Twi-2) to visualize how the bidirectional model fared compared to the unidirectional model under a constant K10 partition protocol.

#### 2.4.2. Experiment 2: SDL Models vs. EDL Models

The aim of this experiment was to determine whether the EDL models are superior to their corresponding SDL models, averaged across our four main datasets (SD-Sford-09, DD-Red-14, DD-Kgg-22, and SD-Twi-2). For this experiment, we compared five EDL models and seven SDL models to demonstrate how joining multiple SDL models can improve performance under a constant K10 partition protocol.

#### 2.4.3. Experiment 3: Effect of Training Size on the Performance of SDL/EDL Models

To validate the robustness of the models, we applied four cross-validation protocols, namely, K2, K4, K5, and K10, to vary the training size for each model and evaluate the corresponding performance drop resulting from reducing the training size. We utilized these partition protocols across the four main datasets (SD-Sford-09, DD-Red-14, DD-Kgg-22, and SD-Twi-2) and averaged the results to illustrate how data size affects our model.

#### 2.4.4. Experiment 4: EDL Models without Attention Block vs. EDL Models with Attention Block

The purpose of this experiment was to observe the change in performance of the SDL and EDL models when augmented with a self-attention block after the classifier in the architecture, compare them with the original EDL model across all datasets (SD-Sford-09, DD-Red-14, DD-Kgg-22, and SD-Twi-2), and benchmark them under a constant partition protocol.

#### 2.4.5. Experiment 5: Domain Adoption of Ensemble Deep Learning Models in Unseen Paradigm

This experiment was one of the most critical, as it aimed to evaluate the EDL model’s performance when encountering cross-domain data (where a model was trained on one domain and was then applied to test a different domain) using an “unseen test dataset”. Specifically, we trained the model on one dataset and evaluated it on a different dataset with varying domains and semantics to demonstrate its generalization ability. Our model showcased domain adaptation following training on a single domain of sentiment data and an evaluation of its ability to adapt to a new domain by testing its performance on depression data. We were able to transfer knowledge learned from the original domain to a new domain and assess the model’s ability to generalize to different tasks and datasets.

### 2.5. Performance Metrics

The proposed models were estimated using the parameters “true positive (TP)”, “true negative (TN)”, “false positive (FP)”, and “false negative (FN)”, which are defined as follows: If a normal/neutral sentiment input is detected as a normal/neutral sentiment by the depression detection mechanism, then it is identified as true positive (TP). If a depressive/negative sentiment input is detected as a depressive/negative sentiment by the depression detection mechanism, then it is identified as true negative (TN). In the other case, if a depressive/negative input is detected as a normal/neutral sentiment by the mechanism, then it is identified as false positive (FP). Finally, if a normal/neutral input is detected as a depressive/negative sentiment by the mechanism, then it is identified as false negative (FN). Using these parameters, we can derive the following PE parameters: (i) *Accuracy*: This denotes the overall correct predictions out of the total predictions made (Equation (5)). (ii) *Recall* (R): This is the number of correctly predicted positive class predictions made of all the positive members in the dataset (Equation (6)). (iii) *Precision* (P): This is the number of correctly predicted positive class predictions to the total number of classified positive predictions (Equation (7)). (iv) *F1-Score* (*F*): This is defined as the harmonic mean of precision and recall. It is useful for imbalanced datasets (Equation (8)). (v) Finally, the area-under-the-curve (AUC) represents the two-dimensional area underneath the plotted ROC curve.

### 2.6. Mean and Standard Deviation of the Statistics 

In this study, we propose formulations for measuring the overall robustness of the model. To accomplish
{   (5)η=TP+TNTP+FP+FN+TN(6)R=TPTP+FN(7)P=TPTP+FP(8)F=2×P×RP+R
this, we measure six quantities in this section. $\bar{\eta }\left(m\right)$ denotes model m’s accuracy summarized over all D datasets; $\bar{\eta }\left(d\right)$ denotes the robustness of dataset d achieved by summarizing M models; ${\bar{\eta }}_{sys}$ denotes the overall system robustness by averaging the accuracy achieved over M models and D datasets; $\bar{\alpha }\left(m\right)$ denotes model m’s area-under-the-curve summarized over all D datasets; $\bar{\alpha }\left(d\right)$ denotes the robustness of dataset d, achieved by summarizing the area-under-the-curve over M models; and ${\bar{\alpha }}_{sys}$ denotes the overall system robustness by averaging the area-under-the-curve achieved over M models and D datasets. All these formulas were computed in Section 3 using the K10 partition protocol.
{   (9)η¯m, K10=∑d=1 Dη m, d, K10D(10)η¯d, K10=∑m=1M η m, d, K10M(11)η¯sys=∑d=1 D∑m=1M η m, d, K10M×D(12)α¯m, K10=∑d=1 Dα m, d, K10D(13)α¯d, K10=∑m=1M α m, d, K10M(14)α¯sys=∑d=1 D∑m=1M α m, d, K10M×D

## 3. Results

The experimental results of the protocols were obtained by employing four main datasets (SD-Sford-09, DD-Red-14, DD-Kgg-22, and SD-Twi-2) and sixteen (11 + 5) models through the utilization of the TensorFlow framework. The training process was executed using a Tesla P100 GPU. Each result was obtained by conducting ten rounds of training and testing, and subsequently, calculating the mean value.

### 3.1. Unidirectional vs. Bidirectional SDL Models

In this study, we demonstrated that bidirectional models consistently outperform unidirectional DL models of comparable architecture. We tested this hypothesis by training and testing six baseline models, with both bidirectional and unidirectional variations. Across all three variations, the bidirectional models consistently outperformed the unidirectional models, as visualized in Table 7, validating our hypothesis. Specifically, the BiLSTM model achieved the greatest absolute increase in performance compared to the LSTM model, with an increase of 2.65% averaged over all four datasets, with BiRNN giving a 1.80% increase over RNN, and BiGRU giving a 1.47% increase over GRU. This can be attributed to the fact that bidirectional models have the ability to process data from past and future inputs, giving them better comprehension of the sequence and context, which can improve performance in comparison to unidirectional models, which only work in one direction.

### 3.2. SDL Models vs. EDL Models (without Attention)

This experiment demonstrates how EDL models outperform their individual components. We evaluated eleven SDL models and five EDL models, and their performance metrics were averaged over four datasets; the results are presented in Table 8. As shown in the table, the EDL models consistently outperformed their individual components in every case, with a mean increase in performance over the five EDL models of **4.49%**. Moreover, the best increase in EDL model performance was observed in EDL4, which showed a **7.22%** absolute increase over its component, SDL8. These results demonstrate the effectiveness of EDL models in improving the overall performance of DL models. 

This could be attributed to the fact that EDL models are able to leverage the strengths of different models and overcome their weaknesses, leading to improved accuracy and generalization ability. In contrast, SDL models may struggle to capture complex relationships in the data or be prone to overfitting. EDL models address these limitations by combining the predictions of multiple models, each with different strengths and weaknesses, thereby reducing the risk of overfitting and improving the robustness of the model.

**Table 8 diagnostics-13-02092-t008:** Evaluation results of all SDL and EDL models using K10 protocol.

Table 8.						EDL Models						Comparative Analysis		Remarks
SDL Type	SDL Model	Mean Accuracy	Mean Precision	Mean Recall	Mean AUC	EDL Type	EDL Model	Mean Accuracy	Mean Precision	Mean Recall	Mean AUC	Absolute Increase %		Top 3 EDL
SDL1	LSTM	84.12%	83.08%	84.04%	0.8194	EDL1	GRU + CNN-LSTM	87.63%	86.83%	86.99%	0.8616	EDL1 > SDL3	0.77%	
SDL2	BiLSTM	86.35%	85.04%	85.54%	0.8554							EDL1 > SDL7	2.47%	
SDL3	GRU	86.96%	86.43%	84.90%	0.864	EDL2	BiSTM + CNN-LSTM	88.61%	87.65%	86.75%	0.8727	EDL2 > SDL2	2.62%	
SDL4	BiGRU	88.24%	87.55%	88.67%	0.876							EDL2 > SDL7	3.62%	
SDL5	RNN	85.49%	86.32%	85.37%	0.8427	EDL3	BERT + CNN-BiLSTM	92.54%	91.81%	92.56%	0.9082	EDL3 > SDL9	6.17%	
SDL6	BiRNN	88.04%	87.38%	88.77%	0.878							**EDL3 > SDL8**	**6.21% ^3^**	Best 3 EDL
SDL7	CNN-LSTM	85.52%	85.57%	85.42%	0.8413	EDL4	ALBERT + CNN-BiLSTM	93.42%	92.74%	93.32%	0.8971	**EDL4 > SDL8**	**7.22% ^1^**	Best 1 EDL
SDL8	CNN-BiLSTM	87.13%	87.51%	87.51%	0.8674							EDL4 > SDL10	5.33%	
SDL9	BERT	87.16%	87.13%	86.53%	0.8608	EDL5	ALBERT + BERT-BiLSTM	95.01%	94.14%	94.11%	0.9313	**EDL5 > SDL10**	**7.13% ^2^**	Best 2 EDL
SDL10	ALBERT	88.69%	88.91%	88.51%	0.876							EDL5 > SDL11	3.36%	
SDL11	BERT-BiLSTM	91.92%	92.04%	91.35%	0.9024							**Mean Increase**	**4.49% ^4^**	

^1^ First Best performing EDL Model (EDL4) as compared to its SDL component (SDL8). ^2^ Second Best performing EDL Model (EDL5) as compared to its SDL component (SDL10). ^3^ Third Best performing EDL Model (EDL3) as compared to its SDL component (SDL8). ^4^ Mean increase in accuracy of EDL Modes as compared to their SDL components.

### 3.3. Cross-Validation Protocols of All Models 

In this experiment, we studied the effect of training data size on the performance of our models. Table 9 and Table 10 present the results of our analysis, showcasing how the accuracy and area-under-the-curve (AUC) metrics, respectively, gradually decrease over different cross-validation protocols (K10 (default), K5, K4, and K2). In this case, the accuracy of EDL5 dropped from **95.01%** when using the K10 protocol to **90.07%** when using the K2 protocol, and the AUC fell from **0.9251** when using the K10 protocol to **0.8667** when using the K2 protocol. Even with a reduced amount of training data in the K2 (50:50) validation protocol, the metrics of our EDL models did not drop significantly, demonstrating the generalizability of the models. These results suggest that our EDL models can be used effectively even when the amount of available training data is limited.

### 3.4. Effect of Attention on the SDL and EDL Models and Its Benchmarking against SemEval Dataset 

The fourth experiment aimed to demonstrate the effect of using an attention layer in all of the SDL and EDL models. For this purpose, we implemented a self-attention channel on top of the EDL models. As demonstrated by Table 11, the use of attention increased the performance of the models summarized over all four main datasets (SD-Sford-09, DD-Red-14, DD-Kgg-22, and SD-Twi-2). According to Table 11, Mean *ae*SDL > *ane*SDL for all five of the PE metrics, and similarly, the mean accuracy of *ae*EDL > *ane*EDL for all five of the PE metrics. This further proves our hypothesis that “attention blocks” are a powerful paradigm in depression and sentimental analysis.

Furthermore, with this experiment, we were able to establish a benchmark on the SemEval 2016 Subtask A dataset, with an accuracy of **85.09%** and an AUC score of **0.8008**; this is the highest accuracy achieved so far using EDL5 with the self-attention block, giving an boost in accuracy of **3.86%**, compared to the best score for SDL11 with the self-attention block. These results, shown in Table 12 and Table 13, demonstrate the effectiveness of our approach in achieving state-of-the-art performance in sentiment analysis on the SemEval dataset.

**Table 11 diagnostics-13-02092-t011:** Effect of attention layer on SDL and EDL models using K10 protocol.

**Performance Metrics**								
**Metrics**		**Mean of Eleven SDL Models**				**Mean of Five EDL Models**		
	**Without** **Attention**	**With** **Attention**	**w/Atten vs.** **w/o Atten**	**Absolute** **Difference**	**Without** **Attention**	**With** **Attention**	**w/Atten vs.** **w/o Atten**	**Absolute** **Difference**
Accuracy	87.24%	89.49%	*ae*SDL > *ane*SDL	2.58%	91.44%	93.97%	*ae*EDL > *ane*EDL	2.76%
Precision	87.00%	88.32%	*ae*SDL > *ane*SDL	1.52%	90.63%	92.15%	*ae*EDL > *ane*EDL	1.67%
Recall	86.96%	89.09%	*ae*SDL > *ane*SDL	2.45%	90.75%	92.86%	*ae*EDL > *ane*EDL	2.33%
F1-Score	86.98%	88.70%	*ae*SDL > *ane*SDL	1.98%	90.69%	92.50%	*ae*EDL > *ane*EDL	2.00%
AUC	0.8621	0.877	*ae*SDL > *ane*SDL	1.73%	0.89412	0.9192	*ae*EDL > *ane*EDL	2.80%
**Comparative Analysis of Attention Layer**								
**Mean Accuracy Comparison**		**Absolute Difference**		**Mean Accuracy Comparison**			**Absolute Difference**	
*ae*SDL > *ane*SDL		2.58%		*ae*SDL > *ane*SDL			1.73%	
*ae*EDL > *ane*EDL		2.76%		*ae*EDL > *ane*EDL			2.80%	
*ane*EDL > *ane*SDL		4.82%		*ane*EDL > *ane*SDL			3.71%	
*ae*EDL > *ae*SDL		5.06%		*ae*EDL > *ae*SDL			4.81%	

**Table 12 diagnostics-13-02092-t012:** Effect of attention on SDL models on benchmark dataset “SemEval” using K10 protocol.

Dataset	SD-SemEval-16 (Without Attention)				SD-SemEval-16 (With Attention)			
Model	Accuracy	Precision	Recall	AUC	Accuracy	Precision	Recall	AUC
LSTM	76.70%	76.11%	76.25%	0.7519	77.04%	76.45%	76.35%	0.754
BiLSTM	78.15%	77.24%	77.82%	0.7599	78.36%	77.40%	77.62%	0.762
GRU	77.16%	76.88%	76.59%	0.7631	77.29%	76.97%	77.21%	0.7645
BiGRU	77.25%	76.96%	77.08%	0.7643	77.34%	77.10%	77.34%	0.7648
RNN	77.11%	76.98%	77.11%	0.7639	77.55%	77.02%	77.24%	0.764
BiRNN	77.26%	77.02%	77.28%	0.7647	77.86%	77.47%	77.33%	0.7655
CNN-LSTM	79.44%	79.20%	79.17%	0.7664	79.91%	79.67%	80.14%	0.7679
CNN-BiLSTM	80.40%	80.29%	80.05%	0.7707	81.42%	80.615	80.22%	0.7709
BERT	81.31%	81.36%	80.95%	0.7744	81.53%	81.40%	80.96%	0.7812
ALBERT	81.73%	80.47%	81.32%	0.7825	81.80%	80.56%	81.43%	0.7832
BERT-BiLSTM	81.74%	80.61%	81.70%	0.7842	81.93%	80.89%	81.81%	0.7897

**Table 13 diagnostics-13-02092-t013:** Effect of attention on EDL models on benchmark dataset “SemEval” using K10 protocol.

Dataset	SD-SemEval-16 (Without Attention)				SD-SemEval-16 (With Attention)			
Model	Accuracy	Precision	Recall	AUC	Accuracy	Precision	Recall	AUC
GRU + CNN-LSTM	79.60%	77.39%	77.74%	0.7716	80.15%	78.03%	78.09%	0.7748
BiLSTM + CNN-LSTM	81.50%	79.67%	79.56%	0.7846	81.19%	80.03%	80.23%	0.7854
BERT + CNN-BiLSTM	81.59%	80.26%	80.35%	0.7896	81.70%	80.44%	80.62%	0.7921
ALBERT + CNN-BiLSTM	82.07%	80.64%	80.53%	0.7978	82.26%	81.05%	80.89%	0.7982
ALBERT + BERT-BiLSTM	82.61%	80.87%	80.60%	0.7998	85.09%	81.04%	81.24%	0.8008

Difference between With Attention and Without Attention in Accuracy for SDL model is 3% (ALBERT + BERT-BiLSTM).

### 3.5. Unseen Tests Using Cross-Domain Testing for SDL and EDL Models 

In this experiment, we demonstrate our model’s ability to perform in a cross-domain setting by conducting unseen tests. We performed 12 sub-experiments on four datasets, involving training on one dataset and testing on a different one, covering all possible combinations. The performance results were averaged out for all the datasets, and the accuracy and percentage differences in seen accuracy and unseen accuracy are shown in Table 14 and Table 15 for the SDL and EDL models. Our analysis showed that the mean difference between unseen and seen accuracy for the SDL models was **~3%**. Similarly, the mean difference between unseen and seen accuracy for the EDL models was **~2.7%**. 

The corresponding AUC and percentage differences are shown in Table 16 and Table 17 for the SDL and EDL models, respectively. Our analysis showed that the mean difference between the unseen and seen AUC for the SDL models was **~3%**. Similarly, the mean difference between the unseen and seen AUC for the EDL models was **~2.4%**. Note that the criterion for a robust design, leading to superior generalizability, was that the difference between seen and unseen analysis be less than **3%** to **5%** [54,55,56]; our system design demonstrates results less than **3%**, which qualifies it as a robust, generalizable, and stable design, which is also part of our running hypothesis.

## 4. Performance Evaluation and Explainable AI

As part of the performance evaluation, the classifiers of the models were determined through their ROC curves, and bar charts were plotted to visualize the performance of the models. ROC curves and bar charts provide a visual representation of the model’s performance. Overall, the performance evaluation provides insight into the strengths and weaknesses of the system and helps to identify areas for improvement. The reliability of the system was assessed to determine its robustness and the stability of the model. This was achieved through various statistical tests, such as the R-squared test (adjusted), and paired *t*-test. The statistical tests were used to determine whether the differences in performance between the models are significant.

As part of the increasing interpretability of the AI models, explainable AI techniques were employed. These techniques provide insights into how the black-box models make decisions and help understand the factors contributing to depression detection.

### 4.1. Receiver Operating Curves

ROC curves are used to evaluate the performance of the models across their entire been operating range. In Figure 7, we visualize the effect of the size of the training data on the EDL5 model by implementing cross-validation protocols K10, K5, K4, and K2. We observe that the AUC for K10 is **0.9251**, and the AUC for K2 is **0.8867**. The ROC performance of the five EDL models (EDL1, EDL2, EDL3, EDL4, and EDL5) is visualized in Figure 8, with EDL5 having the highest AUC score of **0.9251** and EDL1 having the lowest AUC score of **0.8616**.

The bar charts are helpful in visualizing the information present in tables more efficiently. Figure 9 showcases the accuracy of all EDL models averaged over the four datasets, with EDL1 having an accuracy of **87.63%** and EDL5 having an accuracy of **95.01%**. Figure 10 visualizes the effect of accuracy with the change in the amount of training data for EDL5 through the use of cross-validation protocols K2, K4, K5, and K10. The accuracy in K2 drops to **90.07%** from **95.01%** in K10 (default).

### 4.2. Reliability Analysis Using Statistical Tests

The stability of the system was validated through four statistical tests conducted on the EDL models across all five datasets. The tests performed were the adjusted R-squared test, two-tailed Z test, paired *t*-test, and ANOVA test. These tests were conducted to determine whether the predicted data were significant and to monitor the *p*-value in the paired *t*-test and ANOVA test to check whether it was less than 0.01 (*p* < 0.001). The results of these tests are presented in Table 18, across all five EDL models and the five datasets (SD-Sford 09, DD-Red-14, DD-Kgg-22, SD-Twi-2, and SD-SemEval-16).

Along the lines of [57], we conducted these tests and observed that all five EDL models showcased *p* < 0.001 in the paired *t*-test and the ANOVA test, signifying the significance of the data and validating their clinical importance. The adjusted R-squared test, which portrays the correctness of the model, illustrates the extent of a feature’s variance, and the Z in two-tailed tests denotes the Z-score, which describes the standard deviation above or below the mean population.

**Table 18 diagnostics-13-02092-t018:** Statistical tests on EDL models using K10 protocol. EDL1 (GRU+CNN-LSTM); EDL2 (BiLSTM + CNN-LSTM); EDL3 (BERT + CNN-BiLSTM); EDL4 (ALBERT + CNN-BiLSTM); EDL5 (ALBERT + BERT-BiLSTM).

Dataset	Tests	Models				
		EDL1	EDL2	EDL3	EDL4	EDL5
SD-Sford-09	Adjusted R2	0.714	0.789	0.784	0.864	0.879
	Z (two-tailed)	1.37	6.52	−2.56	−1.25	−1.22
	Paired *t*-test	*p* < 0.0005	*p* < 0.0005	*p* < 0.0001	*p* < 0.0001	*p* < 0.0002
	ANOVA test	*p* < 0.0001	*p* < 0.0002	*p* < 0.001	*p* < 0.0002	*p* < 0.0001
SD-SemEval-16	Adjusted R2	0.659	0.714	0.744	0.723	0.7892
	Z (two-tailed)	1.31	−0.679	0.3	2.33	0.7
	Paired *t*-test	*p* < 0.0001	*p* < 0.0001	*p* < 0.0002	*p* < 0.0001	*p* < 0.0001
	ANOVA test	*p* < 0.001	*p* < 0.0003	*p* < 0.0002	*p* < 0.001	*p* < 0.001
DD-Red-14	Adjusted R2	0.735	0.712	0.882	0.825	0.813
	Z (two-tailed)	−6.95	−2.32	−1.02	5.43	1.64
	Paired *t*-test	*p* < 0.0002	*p* < 0.002	*p* < 0.0001	*p* < 0.0001	*p* < 0.0003
	ANOVA test	*p* < 0.0001	*p* < 0.0003	*p* < 0.0002	*p* < 0.0004	*p* < 0.0001
DD-Kgg-22	Adjusted R2	0.874	0.742	0.755	0.713	0.819
	Z (two-tailed)	4.64	0.87	4.45	−0.42	1.43
	Paired *t*-test	*p* < 0.0008	*p* < 0.001	*p* < 0.001	*p* < 0.0005	*p* < 0.0001
	ANOVA test	*p* < 0.0001	*p* < 0.0001	*p* < 0.0002	*p* < 0.0001	*p* < 0.0001
SD-Twi-23	Adjusted R2	0.755	0.736	0.808	0.922	0.932
	Z (two-tailed)	−2.04	−3.28	−0.98	1.54	2.04
	Paired *t*-test	*p* < 0.0001	*p* < 0.0002	*p* < 0.001	*p* < 0.0001	*p* < 0.001
	ANOVA test	*p* < 0.0001	*p* < 0.0002	*p* < 0.0001	*p* < 0.001	*p* < 0.0001

### 4.3. Reliability Analysis Using Statistical Tests

Given that deep learning models, such as BERT, are often considered black box models, we recognized the importance of providing insights into the interpretability of our results. To address this, we employed the “SequenceClassificationExplainer” module from the “transformers-interpret” library in our paradigm, as showcased in Figure 11. This explainer allowed us to calculate the attribution of each word in a given sentence after cleaning, tokenization, and prediction. It enabled us to identify the most impactful tokens contributing to the sentiment classification. Additionally, by using fixed thresholds, we constructed masked sentences that highlight the most impactful tokens.

In our study, a value closer to 0 indicates a depressive sentiment, while a value closer to 1 indicates a non-depressive sentiment. By incorporating this explainable AI technique, our aim was to shed light on the underlying factors influencing sentiment classification. The results over the two datasets are demonstrated in Figure 12 and Figure 13. Although BERT itself does not inherently provide specific interpretability features, leveraging the explainability module helped us address the lack of fixed features and provided additional insights into the decision-making process of our model.

**Figure 11 diagnostics-13-02092-f011:**
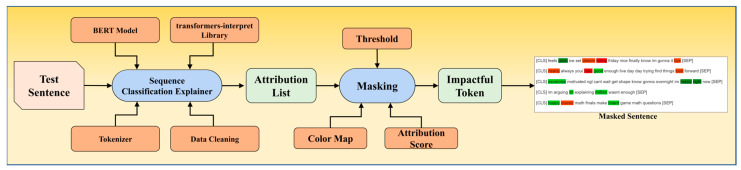
Flowchart of explainable AI module.

**Figure 12 diagnostics-13-02092-f012:**
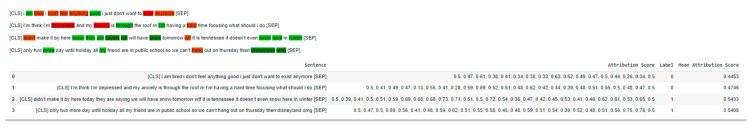
Word attribution using explainable AI on first dataset.

**Figure 13 diagnostics-13-02092-f013:**
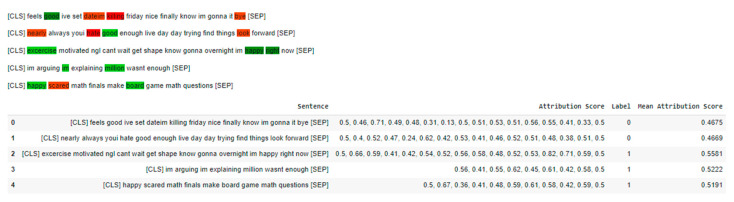
Word attribution using explainable AI on second dataset.

## 5. Discussion

Since we implemented 11 SDL and 5 EDL models on four datasets with and without attention paradigms (16 × 2 = 32 models), we summarize the primary and secondary findings of our comprehensive analysis. Further, it is critical to benchmark our design (*ae*EDL and *ae*SDL) against the existing studies in the domains of depression and sentimental analysis. Another important component is to elaborate on the bounds of the attention mechanism under which it was adapted. Lastly, as part of the discussion, we illustrate the strengths and weaknesses and possible extensions of this study.

### 5.1. Principal Findings

Through an exhaustive study, we have proven *three major* and *three minor* hypotheses. We developed eleven SDL models and five EDL models. Using our four main datasets (SD-Sford-09, DD-Red-14, DD-Kgg-22, and SD-Twi-2), we discovered that *bidirectional* SDL models outperformed *unidirectional* models. Building upon this finding, we discovered that EDL models outperformed their component SDL models by **4.49%**, and yielded better results when they were utilized together in the architecture. Using self-attention layers, we observed significant improvement in the performance of DL models. This was further enhanced by incorporating attention mechanisms into our EDL architecture, leading to benchmark accuracy on the SemEval-2016 dataset. We observed that the increase in the mean accuracy (AUC) of *ae*SDL over *ane*SDL was **2.58% (1.73%)**, and the increase in the mean accuracy (AUC) of *ae*EDL over *ane*EDL was **2.76% (2.80%)**. When comparing EDL vs. SDL for non-attention and attention, the mean *ane*EDL was greater than *ane*SDL by **4.82% (3.71%)**, and the mean *ae*EDL was greater than *ae*SDL by **5.06% (4.81%)**. On benchmarking dataset (SemEval), the best-performing *ae*EDL model (ALBERT + BERT-BiLSTM) was superior to the best *ae*SDL (BERT-BiLSTM) model by **3.86%.** Furthermore, we validated our models through statistical tests, demonstrating their ability to effectively handle cross-domain challenges by performing well on unseen paradigms and predicting on different domains to those on which they were trained. We met the regulatory requirement by showing that the accuracy and AUC differences between unseen and seen paradigms were less than **3%.**

### 5.2. Benchmarking: A Comparative Analysis

The crux of our study was positioned using an *attention-enabled* paradigm in EDL models. These EDL models were designed by fusing DL-based models. Thus, it is important to evaluate our framework against the previous SDL and EDL models. We therefore decided to squarely address the benchmarking efforts in two consecutive steps, with step one involving a comparison of our proposed models with previous DL models and step two consisting of a deeper comparison of our proposed models with previous EDL models. Since the total number of studies in the sentiment analysis and depression detection were 9 and 18, respectively, we organized our benchmarking into two clusters in the form of two tables, namely, Table 19 and Table 20. Table 19 focuses on nine studies that did not use attention in their architecture, and Table 20 consists of studies where attention blocks were an integral part of the paradigm. Table 19 showcases fourteen attributes for each of the nine studies. Columns C1 to C16 are as follows: the year of the study (C1); the last name of the author (C2); the main objective of the paper (C3); the base model (C4); the use of an SDL vs. an EDL model (C5); the fusion or stacking technique used, if any (C6); the main method used (C7); the data type used (C8); data size (C9), the evaluation metric (C10) and evaluation score (C11); scientific validation (C12) and clinical validation (C13); and the conduction of an unseen paradigm, if any (C14). 

Out of the nine studies, six studies [58,59,60,61,62,63] worked on sentiment analysis, while two studies [64,65] carried out work on depression detection. Study [66] approached suicide detection as their objective. LSTM and BiLSTM, as SDL or as HDL, were the main approaches used in studies [58,60,62,63], while studies [59,64] used statistical ML algorithms, and the authors of [65] implemented decision trees with pruning for classification. Four out of the nine studies [58,60,61,62,63] used SDL models (columns C5, C6). In contrast, the authors of [64,65,66] used EDL models as the main architecture through the use of stacking, feature extraction, or max voting.

The studies demonstrated up to four kinds of data source, namely, social media chat, reviews, or published psychological data (column C8). Unlike these, in our proposed study (R10), keeping generalizability in mind, we used five kinds of data source, namely, Twitter, Twitter (Stanford), Reddit, Kaggle, and SemEval (for benchmarking). Furthermore, these nine studies had a data size range of 718 to 550,000 sentences (column C9), while in our proposed study (R10), the data size ranged from 10,000 to 1,000,000 sentences. All of these studies computed at least one out of accuracy, F1-Score, or AUC (columns C10, C11). Our study (R10) outperformed existing studies and yielded a mean accuracy of **91.44%** across all EDL models and **95.01**% in the best EDL5 model, and an F1-Score of **0.8941**. Seven out of the nine studies [58,59,60,62,63,64,65] presented some sort of scientific or clinical validation by performing an ablation study, providing *p*-values of less than 0.001, or by performing cross-validation (columns C12, C13), unlike in our proposed study (R19), where we conducted exhaustive tests, including six individual statistical tests that yielded *p*-values of less than 0.001, deployed four cross-validation protocols, and achieved an overall standard deviation of less than **2.5%**. Further, it is noted that only our proposed work (R10) conducted a true unseen paradigm (column C14) by training and testing the model on datasets of different domains, thus proving its generalizability over cross-domains.

Table 20 shows the state-of-the-art DL models used for sentiment analysis and depression detection. It showcases fifteen attributes for each of the eighteen studies: Columns C1 to C15 are as follows: the year of the study (C1); last name of the author (C2); the main objective of the paper (C3); the base model (C4); the use of an SDL vs. an EDL model (C5); the fusion or stacking technique used, if any (C6); the attention block technique (C7); the main method used (C8); the data type used (C9); data size (C10); evaluation metric (C11) and evaluation Score (C12); scientific validation (C13) and clinical validation (C14); and the conduction of an unseen paradigm, if any (C15).

Most of the studies worked on sentiment analysis [67,68,69,70,71,72,73,74,75,76,77,78,79,80], while study [81] worked on stance detection, study [82] focused on emotion recognition, study [83] focused on bias identification, and finally, study [84] focused on depression detection (Column C3). Unlike their singular focuses, our study’s paradigm (R19) was domain adaption; hence, our objective was threefold—sentimental analysis, and depression and suicide detection. Studies [67,68,70,71,72,73,74,75,76,77,78,80,81,83,85] used a single model as the base classifier (column C4), while only four studies [69,79,82,84] used HDL models as the base model. Notably, to outperform the existing results, our proposed study (R19) squarely used a hybrid of ALBERT, BERT, and BiLSTM as one of the base EDL models.

Ten out of the eighteen studies [67,69,72,73,74,76,77,80,83,84] used SDL models (columns C5, C6). In contrast, the authors of [68,70,71,75,78,79,81,82] and our proposed study (R19) used EDL models as the main architecture, using concatenation, fusion, or weighted sums. All of the studies [67,68,69,70,71,72,73,74,75,76,77,78,79,80,81,82,83,84] demonstrated three to four kinds of data source, namely, social media chat, reviews, or public communications (column C9). Unlike these, in our proposed study (R19), keeping generalizability in mind, we used five kinds of data source, namely, Twitter, Twitter (Stanford), Reddit, Kaggle, and SemEval (benchmarking). Furthermore, these eighteen studies, apart from [63], had a data size range of 1250 to 100,000 sentences (column C10), while in our proposed study (R19), the data size ranged from 17,719 to 1,600,000 sentences. Except for study [83], the remaining seventeen studies [67,68,69,70,71,72,73,74,75,76,77,78,79,80,81,82,84] computed at least one out of accuracy, F1-Score, or AUC (columns C11, C12). Our study (R19) outperformed the existing studies and yielded an accuracy of **93.97%** and an F1-Score of **0.9192**. Furthermore, our study provided benchmark accuracy over the benchmark public SemEval-2016 (*SD-SemEval-16*) dataset, achieving **85.09%** accuracy on our *ae*EDL model (ALBERT + BERT-BiLSTM).

Studies [67,68,69,71,74,76,78,79,80,82,83] presented some sort of scientific and clinical validation by performing ablation studies, yielding *p*-values of less than 0.001, or performing cross-validation (columns C13, C14), unlike in our proposed study (R19); we conducted six individual statistical tests that yielded *p*-values of less than 0.001, deployed four cross-validation protocols, and achieved an overall standard deviation of less than **2.5%**. While only three studies attempted to validate unseen data [74,83,84] by employing sub-sampling, data merging, or cross-domain validation (column C15), it is noted that only our proposed work (R19) conducted a true unseen paradigm through training and testing on data of different domains, thus proving its generalizability.

**Table 19 diagnostics-13-02092-t019:** A table showing the studies carried out that did not use attention blocks.

	C1	C2	C3	C4	C5	C6	C7	C8	C9	C10	C11	C12	C13	C14
	Yr.	Last Name	Objective (Analysis)	AI Models	SDL vs. EDL	EDL Type	Method	Data Type	Data Size	Eval. Metric	Score	Sci. Val.	Clin. Val.	Unseen Val.
**R1**	2020	Zhu et al. [63]	Sentiment Analysis	CNN-BiLSTM	SDL	SDL	Context Vector via Kernel Optimization Function	Movie Review Text Data	11,846	Accuracy	85.4	X	K5	X
**R2**	2020	Yilmaz et al. [62]	Multi-Label Sentiment Analysis	BiLSTM	SDL	Dynamic Weighting	Multi-Label Classification using Balancing of classes	Text in 100 Languages—Mixed Type	X	Macro F-Score Micro F-Score	0.584 0.696	Ablation	K7	✔.
**R3**	2020	Bibi et al. [59]	Sentiment Analysis	Clusters of Naive Bayes and SVM	EDL	Clustering—Max Voting	Hierarchical Clustering ML-based Model—Max Voting	4 Datasets (Tweet)	3844/4578/718/3000	AUC Accuracy	0.74 75	X	K10	X
**R4**	2021	Behera et al. [58]	Sentiment Analysis	CNN-LSTM	SDL	SDL	Convolution LSTM Hybrid	4 Text Datasets (Reviews)	25,000/13871	F-Score Accuracy	0.8302 83.13	*p* < 0.05	K5, K10	X
**R5**	2022	Tong et al. [65]	Depression Detection	Decision Trees	EDL	Boosting and Pruning	Boosting Pruning Ensemble Model	3 Datasets (Tweet)—CLPsych	2558/5304/58,810	F-Score Accuracy	0.9246 90.9	Ablation	SD < 2.5	X
**R6**	2022	J. Kumar et al. [60]	Multi-Label Sentiment Analysis	LSTM + GRU	SDL	SDL	Gender-Based Multi-Aspect Sentiment Detection	Text (Reviews)	1722/2438	Accuracy	83.58	X	K5	X
**R7**	2022	Li et al. [66]	Suicide Detection	CNN	EDL	Hierarchical Ensemble	Hierarchical Ensemble Model for Imbalanced Data	2 Datasets (Weibo)	550,000/7329	F-Score Accuracy	0.8962 93.11	Imbalance	X	X
**R8**	2022	Liu et al. [64]	Depression Detection	KNN, NB, LG1, LG2	EDL	Stacking of Base Classifiers	Hybrid Feature Selection and Ensemble Model	Blog Text (Weibo)	X	Accuracy	90.27	X	K10	X
**R9**	2023	Wu et al. [61]	Sentiment Analysis	Graph Network Model	SDL	SDL	Graph Knowledge-Aware Graph Network	4 Text Datasets	6.940/4.728/2.966/12,522	F-Score Accuracy	0.8194 87.01	X	X	X
**R10**	2023	Singh et al. (This paper)	Sentiment/Depression Analysis	ALBERT-CNN-BiLSTM	EDL	Concatenation of Solo Models	Domain-Adaptive Ensemble Models for Depression Detection	Text (Tweet) (Reddit) (Kaggle)	1,600,000/26,000/27,977/31,000/17,719	F-Score Accuracy	0.8941 91.44 **(95.01) ^1^**	SD < 2.5; *p* < 0.001	Cross-Validation	✔

^1^ Best Accuracy achieved by EDL5 Model (ALBERT + BERT-BiLSTM).

**Table 20 diagnostics-13-02092-t020:** A table showing the studies carried out that used attention blocks.

Table 10. *Cont.*	C1	C2	C3	C4	C5	C6	C7	C8	C9	C10	C11	C12	C13	C14	C15
	Yr.	Last Name	Objective (Analysis)	AI Models	SDL vs. EDL	EDL Type	Attention Type	Method	Data Type	Data Size	Eval. Metric	Score	Sci. Val.	Clin. Val.	Unseen Val.
**R1**	2019	Long et al. [67]	Sentiment Analysis	LSTM	**SDL**	SDL	Cognition-Grounded Attention	Cognition-Grounded Attention for improving LSTM Model	3 Text Datasets (Reviews)	84,919/1631/12,836	RMSE Accuracy	0.685 65.5	Ablation	**X**	**X**
**R2**	2020	Zhai et al. [70]	Sentiment Analysis	BiLSTM	EDL	Attention Fusion	Multi-Attention	Fusion Modeling for Educational Data	3 Text Datasets	5052	Accuracy	79.6	**X**	**X**	**X**
**R3**	2020	Zhang et al. [71]	Sentiment Analysis	BiLSTM	EDL	Weighted Sum/Capsules	Capsule Attention	Knowledge-Guided Network using Capsule Attention	5 Text Datasets	6940	Macro F-Score Accuracy	0.7072 88.47	*p* < 0.001	Tests Performed	**X**
**R4**	2020	Yang et al. [69]	Sentiment Analysis	CNN-GRU	**SDL**	SDL	Attention	Lexicon Analysis using Attention	Text (E-Commerce)	100,000	Accuracy	93.2	Ablation	K Cross	**X**
**R5**	2021	Wang et al. [74]	Aspect and Sentiment Analysis	BERT	**SDL**	SDL	Attention	Aspects, Sentiment and opinion Extraction	4 Text Datasets	5971	F-Score Accuracy	0.630 91.7	Ablation	4	✔ Sub-Sample
**R6**	2021	Zhang et al. [76]	Sentiment Analysis	LSTM	**SDL**	SDL	Influence Attention	Interactive LSTM	Text Conversations	24,072	F-Score Accuracy	0.779 78.0	Credibility LSTM	**X**	**X**
**R7**	2021	Su et al. [73]	Sentiment Analysis	BERT	**SDL**	SDL	BERT Self-Attention	BERT-Based Self-Attention	3 Text Datasets	1014/2889/1741	Macro F-Score Accuracy	0.8234 87.86	**X**	**X**	**X**
**R8**	2021	Al-Ghadir et al. [81]	Stance Detection	ML	EDL	Fusion of ranked lists	Fusion Attention	Weighted KNN (ML Models Only)	Text (Tweet)	4163	Macro F-Score	0.7645	**X**	**X**	**X**
**R9**	2022	Tu et al. [78]	Sentiment Classification	Transformers	EDL	Dialogue Transformer	Attention	Attention Context- and Sentiment-Aware Networks	Text Data (Emotion)	13,708/103,607/9489	Accuracy	58.31	Ablation	**X**	**X**
**R10**	2022	P. Kumar et al. [82]	Text Emotion Recognition	BERT BiLSTM	EDL	Ensembled Model with Concatenation	Self-Attention	Dual Channel Module Concatenation Attention	4 DS News/Text/Tweets	7665/5025/1250/2000	Accuracy	79.17	K3, K5, K10	Different Datasets	**X**
**R11**	2022	Mei et al. [77]	Sentiment Classification	RNN	**SDL**	SDL	Attention	Task-Aware Dropout for RNN	Continual Sentiment data	24 × 5000	Accuracy	85.52	**X**	**X**	**X**
**R12**	2022	Huang et al. [72]	Sentiment Analysis	LSTM	**SDL**	SDL	Influence Attention	Enhanced LSTM using Emotion Estimator	4 Text Datasets (English and Chinese)	5000/5000/15,000/15,000	Accuracy	81.15	**X**	Inter-Class Val.	**X**
**R13**	2022	Liu et al. [83]	Political Bias	GPT-2	**SDL**	SDL	Attention	Language Models for bias detection	Text (Media Outlets)	260,000	Bias	0.339	*p* < 0.001	*p* < 0.001	✔ Merging
**R14**	2023	Mosin et al. [80]	Sentiment Analysis	Transformers	**SDL**	SDL	Vocabulary Transfer	Fine-Tuning Transformers	4 Text Datasets (Question–Answer)	201,000/1,300,000/3,688,358	Accuracy	83.1	Token Shuffle	**X**	**X**
**R15**	2020	Yang et al. [68]	Sentiment Analysis	Gated CNN	EDL	Fusion	Self-Attention	Aspect-based Sentimental Attention using Gated CNN	2 Text Datasets (Reviews)	6000/5000	Accuracy	81.4	K10	**X**	**X**
**R16**	2022	Zhang et al. [75]	Sentiment Analysis	Transformers	EDL	Concatenation with Enhancement Nodes	Multi-Task	MultiTask Transformer Network using Mapping	2 Text Datasets (Twitter, Reviews)	69,672	F-Score Accuracy	0.778 77.4	**X**	**X**	**X**
**R17**	2022	Lu et al. [79]	Sentiment Analysis	BERT— BiLSTM	EDL	Attention Fusion	Attention Scores	Graph Attention Network using Dual Channel Edges	6 Text Datasets (Reviews)	12,522/2343/4688/2966	F-Score Accuracy	0.7676 80.56	Ablation	**X**	**X**
**R18**	2022	Nadeem et al. [84]	Depression Detection	GRU—LSTM	**SDL**	SDL	Self-Attention	Hybrid Ternary Classification using Self-Attention	Text (Tweet)	31,000	F-Score Accuracy	0.829 82.9	**X**	**X**	✔ Cross-Domain Val.
**R19**	2023	Singh et al. (This paper)	Sentiment/Depression Analysis	ALBERT-CNN- BiLSTM	EDL	Concatenation and Self-Attention	Self-Attention	Domain Adaptive Ensemble Models with self-attention	5 Text Datasets (Tweet) (Reddit) (Kaggle)	1,600,000/26,000/27,977/31,000/17,719	F-Score Accuracy	0.9192 93.97 (**95.01**) ^1^	SD < 2.5; *p* < 0.001	Cross-Validation	✔
			✔		✔		✔	✔				✔	✔	✔	✔

^1^ Best Accuracy achieved by EDL5 Model (ALBERT + BERT-BiLSTM).

### 5.3. A Special Note on Attention in Depression Detection

Attention mechanisms help in depression detection by allowing the model to selectively focus on important parts of the input text, instead of treating the entire text equally. This is particularly useful in depression detection, where certain words or phrases may be more indicative of depression in some individuals than in others. For example, an attention mechanism could help the model to identify important keywords or phrases that are highly indicative of depression, such as negative self-talk, hopelessness, or social isolation. 

In depression detection, certain words or phrases may carry more weight than others, and attention mechanisms can help the model identify and prioritize these important features. Additionally, attention mechanisms can help the model better understand the context and meaning of the text by focusing on *relevant* information and ignoring *irrelevant* information. This can lead to improved accuracy and performance of the model in identifying depression in text data.

#### Strengths, Weakness, and Extensions

This article focuses on the application of EDL models and attention layer for depression detection. The study shows significant improvement in predicting sentiment and depression from multiple data sources, making the proposed EDL a benchmark in the field of depression detection. The EDL model outperforms existing studies on two datasets. Additionally, cross-validation, clinical validation, and unseen implementations prove the system’s robustness and domain adaptability, as it performs fairly well on a different domain to that on which it was trained, demonstrating its generalization ability.

Due to the limited availability of high-quality open datasets on depression, the existing study focused mainly on training classifiers on specific datasets. Consequently, the model’s accuracy was not improved beyond 95%, although it still outperformed existing studies. To encourage further research in the field and improve current benchmark models, high-quality datasets of substantial size are necessary to build more robust and optimal models. The scope of this approach focuses only on the NLP text-based approach, and hence, we adapted the Twitter and Reddit dataset as they follow the same paradigm. This was an in-depth explanation of the domain adaption paradigm, which focused on the adaption of ensemble-based NLP models through extensive experimentation. We developed 11 solo deep learning models and 5 ensemble models, which were specifically designed to leverage their capabilities in detecting depression from users’ text patterns. Additionally, within our ensemble models, we incorporated attention channels to enhance explainability, highlighting key textual features that contribute to the classification decision. By employing these techniques, we aim to achieve both high performance in depression detection and meaningful explanations for the model’s decisions, ensuring transparency and interpretability in our classification approach.

The exploration of multimodal videos and images will be considered as a potential continuation in future research. Through such research, we can explore datasets from visual-based social media platforms such as TikTok, YouTube, and Instagram. Here, we will shift our focus to video classification, which requires different methodologies compared to the NLP-based classification we have utilized thus far. We will employ computer vision-based classification models to analyze visual cues, facial expressions, body language, and other visual elements in order to detect depression using an entirely different paradigm.

In the future, our goal is to develop new datasets and explore novel architectures, such as Generative Adversarial Networks (GANs), for improving depression detection. We aim to compare these new models, such as the fusion of ML with exhaustive feature space with DL [86], to our existing EDL models to evaluate their performance and perform variability analysis [87]. Additionally, we plan to develop new loss functions and incorporate multiple loss functions into our *ae*EDL models, as adopted in the imaging framework, to increase their robustness and improve their performance metrics [34,88]. Additionally, design systems can be pruned to reduce the size of the training models [89], and artificial intelligence designs are susceptible to bias; we intend to work on understanding studies and rank them according to their bias [90,91,92,93]. 

Lastly, there have been studies in different domains, such as immunology [94,95], cardiovascular risk assessment [96], and psoriasis diagnosis [97], where cloud-based end-to-end systems are used for detection and moderation. We therefore intend to use a similar paradigm to create an automated and scalable cloud-based system using research into AI sentiment analysis to interpret the emotional content present in various forms of communication, such as text messages, social media posts, and online interactions.

The proposed cloud system follows a layered architecture, where the presentation layer operates locally on users’ devices, while the business and persistence layers are hosted on the cloud. This architecture ensures a user-friendly experience by providing real-time sentiment analysis and emotional guidance directly on the device. Additionally, this architecture facilitates secure connectivity between the system and psychologists, enabling them to access and utilize the system’s insights to provide personalized support and assistance to their patients. The automated moderation provided by this system can greatly benefit psychologists in their practice. When patients visit a psychologist, it can sometimes be challenging for them to express their emotions fully. With the assistance of our automated system, psychologists can gain deeper insights into their patients’ emotional well-being. This enhanced understanding will enable psychologists to provide more personalized and tailored treatment plans, improving the effectiveness of their interventions. By utilizing this system, psychologists can leverage technology to follow up with their patients’ mental well-being in a more comprehensive and individualized manner.

The cloud-based nature of our system will play a crucial role in its capabilities. It will enable the secure storage and processing of a vast amount of data, allowing the system to continuously learn and enhance its understanding of emotions. This accumulated knowledge and analysis of sentiment data contribute to a more robust and accurate sentiment analysis process. Furthermore, the integration of this system into mobile phone-based applications will provide users with convenient access to its features. Users can benefit from real-time guidance and support, empowering them to manage their emotions and prioritize their mental well-being more effectively. This sentiment analysis system will act as a personal mental guide in a robust pipeline, operating locally to help users recognize and address their emotions. Additionally, the integration of AI sentiment analysis could enable holistic support for mental well-being, positively impacting individuals’ lives by providing timely assistance and resources based on their emotional needs. 

## 6. Conclusions

Our study presents a novel paradigm for depression detection and sentimental analysis in a cross-domain framework based on text inputs. This utilizes five kinds of attention-enabled ensemble deep learning model designed using eleven kinds of solo deep learning model. A comprehensive data analysis was conducted using four kinds of dataset to prove our hypothesis. Further, a benchmarking strategy was developed on the standardized SemEval dataset, establishing our model’s superior performance both in classification accuracy and area-under-the-curve. As part of a generalizability assessment, “seen” and “unseen” experiments were conducted, with the model meeting the regulatory requirements. Finally, the system’s reliability and stability were demonstrated using clinical tests. 

## Figures and Tables

**Figure 2 diagnostics-13-02092-f002:**
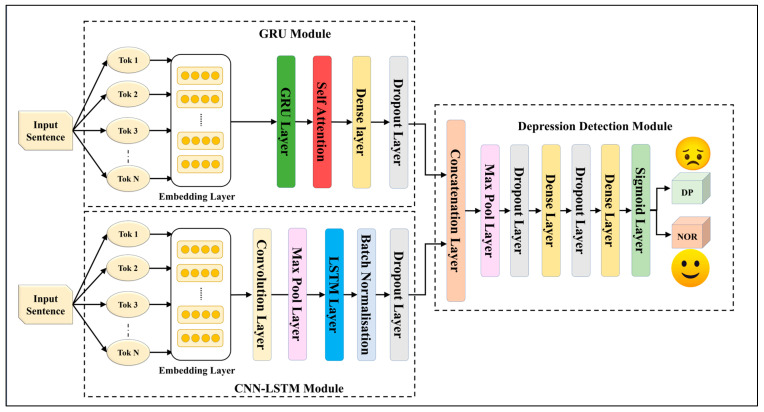
Architecture of EDL1 (SDL3 + SDL7).

**Figure 3 diagnostics-13-02092-f003:**
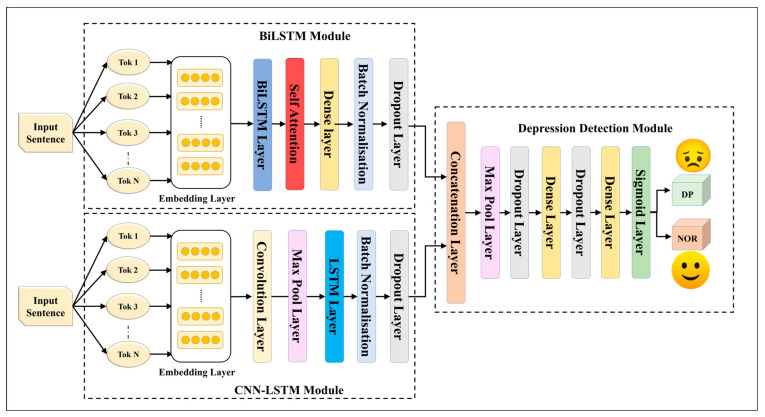
Architecture of EDL2 (SDL2 + SDL7).

**Figure 4 diagnostics-13-02092-f004:**
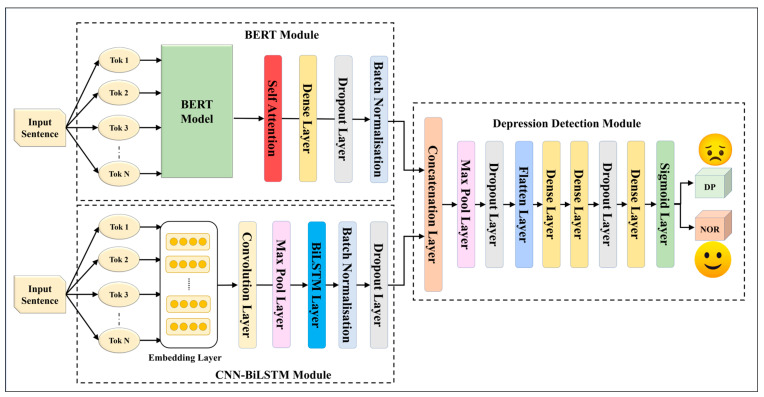
Architecture of EDL3 (SDL8 + SDL9).

**Figure 5 diagnostics-13-02092-f005:**
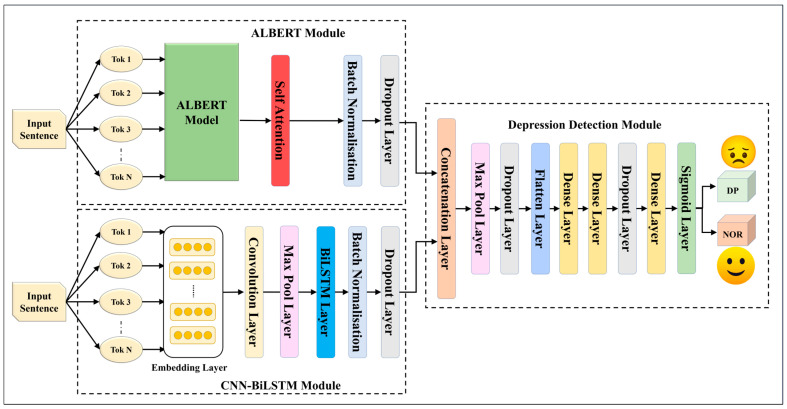
Architecture of EDL4 (SDL8 + SDL10).

**Figure 6 diagnostics-13-02092-f006:**
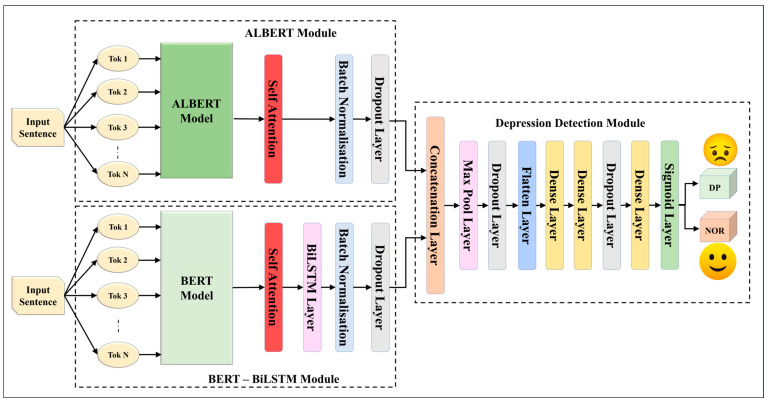
Architecture of EDL5 (SDL10 + SDL11).

**Figure 7 diagnostics-13-02092-f007:**
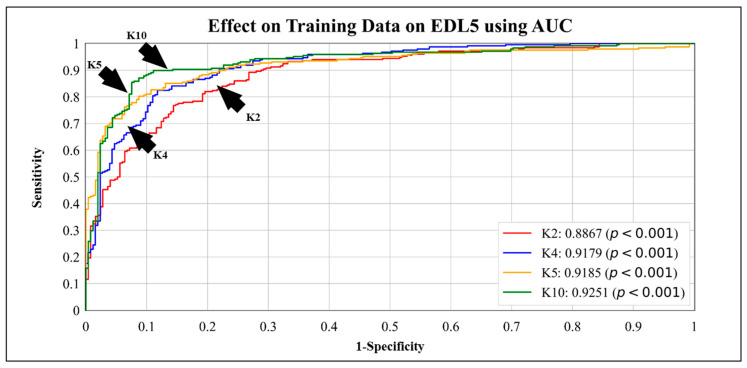
ROC curve of EDL5 (ALBERT + BERT-BiLSTM) using K10, K5, K4, and K2 protocols.

**Figure 8 diagnostics-13-02092-f008:**
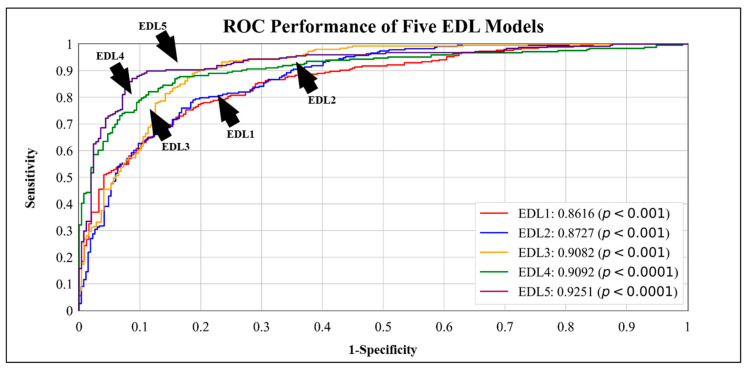
ROC curve of all EDL models using K10 protocol.

**Figure 9 diagnostics-13-02092-f009:**
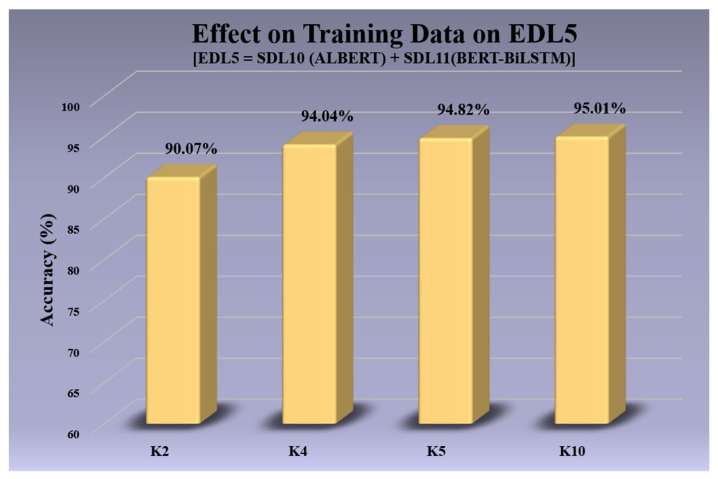
Accuracy of EDL5 (ALBERT + BERT-BiLSTM) using K10, K5, K4, and K2 protocols.

**Figure 10 diagnostics-13-02092-f010:**
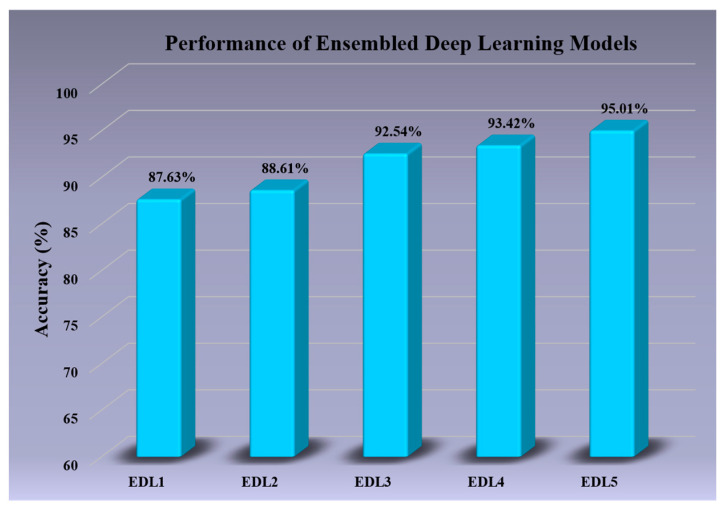
Accuracy of all five EDL models using K10 protocol.

**Table 1 diagnostics-13-02092-t001:** Visual description of “SD-Sford-09” dataset *.

Tweet	Sentiment
Oh dear, my Devilder prediction wasn’t the best…. sorry Irena I only saw the 3rd, Hanescu was pounding the ball and not missing.	0
@switchfoot http://twitpic.com/2y1zl (accessed on 10 January 2023) *—Awww, that’s a bummer. You shoulda got David Carr of Third Day to do it.;D	0
Is sad because the gas station didn’t have french vanilla cappuccino.	0
@rickyzea I be rocking the meds…time to give the old gal her magic pills	4
Orange juice for breakfast this morning, think I’ll splurge at lunch!	4
I have so much to do but I don’t mind one bit because life is just brilliant right now	4

* URL: http://help.sentiment140.com/home (accessed on 10 January 2023).

**Table 2 diagnostics-13-02092-t002:** Visual description of DD-Red-14 dataset.

Reddit Post	Class
i am tired i don’t feel anything good i just don’t want to exist anymore	Depressive
I’m think I’m depressed and my anxiety is through the roof rn I’m having a hard time focusing what should i do	Depressive
didn’t make it by here today they are saying we will have snow tomorrow wtf it is tennessee it doesn’t even snow here in winter	Neutral
only two more day until holiday all my friend are in public school so we can’t hang out on thursday then disneyland omg	Neutral

**Table 3 diagnostics-13-02092-t003:** Visual description of dataset DD-Kgg-22.

Reddit Post	Class
feels good ive set dateim killing friday nice finally know im gonna it bye	1
nearly always youi hate good enough live day day trying find things look forward	1
excercise motivated ngl cant wait get shape know gonna overnight im happy right now	0
im arguing im explaining million wasnt enough	0
happy scared math finals make board game math questions	0
started playing genshin impact yesterday oh god fun id highly recommend it	0

**Table 4 diagnostics-13-02092-t004:** Visual description of SD-SemEval-16 dataset.

Tweet	Class
Come check out @shoutheyband at the black dahlia on saturday night! It’s going to be a party!	Positive
Dethklok, All That Remains, Machine Head, and The Black Dahlia Murder at Stage AE November 8th..HIGHLY considering going to that.#heavymetal	Positive
Talking about ACT’s &amp;&amp; SAT’s, deciding where I want to go to college, applying to colleges and everything about college stresses me out	Negative
The pain is far deeper than a Billy cundiff missed field goal. Gotta wake up and forget about it tomorrow. #Orioles #stayhungry	Negative
Finally starting the 5th season of #Dexter. See ya later, weekend!	Positive

**Table 6 diagnostics-13-02092-t006:** Hyperparameter table for EDL models using K10 protocol.

Hyperparameter Table					
EDL Type	EDL Model	Optr *	CVP	ILR	Max Epochs
EDL1	GRU + CNN-LSTM	Adam	9:1	2.00 × 10^−5^	30
EDL2	BiSTM + CNN-LSTM	Adam	9:1	2.00 × 10^−5^	40
EDL3	BERT + CNN-BiLSTM	SGD	9:1	2.00 × 10^−4^	40
EDL4	ALBERT + CNN-BiLSTM	SGD	9:1	1.00 × 10^−4^	45
EDL5	ALBERT + BERT-BiLSTM	SGD	9:1	1.00 × 10^−4^	50

* Optr: optimizer; CVP: cross-validation partition; ILR: initial learning rate.

**Table 7 diagnostics-13-02092-t007:** Evaluation results of all solo unidirectional and bidirectional deep learning models (K10 protocol).

SDL Type	SDL Model	Mean Accuracy	Mean Precision	Mean Recall	Mean AUC	Comparative Analysis	
						Absolute Increase (%)	
LSTM vs. BiLSTM							
SDL1	LSTM	84.12%	83.08%	84.04%	0.8194	BiLSTM > LSTM	2.65%
SDL2	BiLSTM	86.35%	85.04%	85.54%	0.8554		
GRU vs. BiGRU							
SDL3	GRU	86.96%	86.43%	84.90%	0.864	BiGRU > GRU	1.47%
SDL4	BiGRU	88.24%	87.55%	88.67%	0.876		
RNN vs. BiRNN							
SDL5	RNN	85.49%	86.32%	85.37%	0.8427	BiRNN > RNN	1.80%
SDL6	BiRNN	88.04%	87.38%	88.77%	0.878		

**Table 9 diagnostics-13-02092-t009:** Accuracy metrics of five EDL models using different cross-validation protocols.

Accuracy (%) of Five EDL Models Using Cross-Validation Protocols						
EDL	EDL Models	K2 (50:50)	K4 (75:25)	K5 (80:20)	K10 (90:10)	SD
EDL1	GRU + CNN-LSTM	84.74%	85.51%	86.29%	87.63%	1.07%
EDL2	BiSTM + CNN-LSTM	84.26%	87.33%	88.39%	88.61%	1.74%
EDL3	BERT + CNN-BiLSTM	88.20%	90.89%	92.30%	92.54%	1.73%
EDL4	ALBERT + CNN-BiLSTM	88.59%	92.83%	93.21%	93.42%	1.99%
**EDL5** ^1^	**ALBERT + BERT-BiLSTM**	**90.07%**	**94.04%**	**94.82%**	**95.01%**	**2.0%**

^1^ Best performing EDL Model.

**Table 10 diagnostics-13-02092-t010:** AUC metrics of five EDL models using different cross-validation protocols.

AUC (0–1) of All EDL Models Using Cross-Validation Protocols					
EDL	EDL Models	K2 (50:50)	K4 (75:25)	K5 (80:20)	K10 (90:10)
EDL1	GRU + CNN-LSTM	0.846	0.8503	0.86	0.8616
EDL2	BiSTM + CNN-LSTM	0.8588	0.8651	0.8619	0.8727
EDL3	BERT + CNN-BiLSTM	0.8744	0.8869	0.8925	0.9082
EDL4	ALBERT + CNN-BiLSTM	0.8627	0.8914	0.8965	0.9092
**EDL5** ^1^	**ALBERT + BERT-BiLSTM**	**0.8867**	**0.9179**	**0.9185**	**0.9251**

^1^ Best performing EDL Model.

**Table 14 diagnostics-13-02092-t014:** Accuracy metrics of SDL models on unseen dataset using K10 protocol.

SDL Type	SDL Models	Seen Accuracy (sa)	Unseen Accuracy (ua)	Percentage Difference (sa − ua)/sa × 100
SDL1	LSTM	84.12%	80.24%	4.61%
SDL2	BiLSTM	86.35%	83.41%	3.40%
SDL3	GRU	86.96%	84.77%	2.52%
SDL4	BiGRU	88.24%	85.80%	2.77%
SDL5	RNN	85.49%	83.04%	2.87%
SDL6	BiRNN	88.04%	85.75%	2.60%
SDL7	CNN-LSTM	85.52%	83.15%	2.77%
SDL8	CNN-BiLSTM	87.13%	85.16%	2.26%
SDL9	BERT	87.16%	84.95%	2.53%
SDL10	ALBERT	88.69%	86.93%	1.98%
SDL11	BERT-BiLSTM	91.92%	88.03%	4.23%

Mean difference between unseen and seen accuracy for SDL model is 2.95~3%.

**Table 15 diagnostics-13-02092-t015:** Accuracy metrics of EDL models on unseen dataset using K10 protocol.

EDL Type	EDL Models	Seen Accuracy (sa)	Unseen Accuracy (ua)	Percentage Difference (sa − ua)/sa × 100
EDL1	GRU + CNN-LSTM	87.63%	84.87%	3.15%
EDL2	BiLSTM + CNN-LSTM	88.61%	86.59%	2.27%
EDL3	BERT + CNN-BiLSTM	92.54%	90.45%	2.26%
EDL4	ALBERT + CNN-BiLSTM	93.42%	91.02%	2.57%
EDL5	ALBERT + BERT-BiLSTM	95.01%	91.97%	3.20%

Mean difference between unseen and seen accuracy for EDL model is 2.69~2.7%.

**Table 16 diagnostics-13-02092-t016:** AUC metrics of SDL models on unseen dataset using K10 protocol.

SDL Type	SDL Models	Seen AUC (sa)	Unseen AUC (ua)	Difference (%) (sa − ua)/sa × 100
SDL1	LSTM	0.8194 (*p* < 0.001)	0.7815 (*p* < 0.02)	4.63%
SDL2	BiLSTM	0.8554 (*p* < 0.001)	0.8226 (*p* < 0.002)	3.83%
SDL3	GRU	0.864 (*p* < 0.003)	0.842 (*p* < 0.005)	2.54%
SDL4	BiGRU	0.876 (*p* < 0.002)	0.8529 (*p* < 0.001)	2.64%
SDL5	RNN	0.8427 (*p* < 0.001)	0.8226 (*p* < 0.01)	2.39%
SDL6	BiRNN	0.878 (*p* < 0.001)	0.8581 (*p* < 0.001)	2.27%
SDL7	CNN-LSTM	0.8413 (*p* < 0.001)	0.8225 (*p* < 0.01)	2.23%
SDL8	CNN-BiLSTM	0.8674 (*p* < 0.01)	0.8324 (*p* < 0.01)	4.03%
SDL9	BERT	0.8608 (*p* < 0.001)	0.8421 (*p* < 0.01)	2.17%
SDL10	ALBERT	0.876 (*p* < 0.001)	0.851 (*p* < 0.001)	2.85%
SDL11	BERT-BiLSTM	0.9024 (*p* < 0.0001)	0.8666 (*p* < 0.0001)	3.97%

Mean difference between unseen and seen AUC for SDL model is 3.05~3%.

**Table 17 diagnostics-13-02092-t017:** AUC metrics of EDL models on unseen dataset using K10 protocol.

EDL Type	EDL Models	Seen AUC (sa)	Unseen AUC (ua)	Difference (%) (sa − ua)/sa × 100
EDL1	GRU + CNN-LSTM	0.8616 (*p* < 0.002)	0.8432 (*p* < 0.001)	2.14%
EDL2	BiLSTM + CNN-LSTM	0.8727 (*p* < 0.0001)	0.8526 (*p* < 0.001)	2.30%
EDL3	BERT + CNN-BiLSTM	0.9082 (*p* < 0.0001)	0.8862 (*p* < 0.001)	2.42%
EDL4	ALBERT + CNN-BiLSTM	0.9092 (*p* < 0.001)	0.8895 (*p* < 0.001)	2.17%
EDL5	ALBERT + BERT-BiLSTM	0.9251 (*p* < 0.0001)	0.9009 (*p* < 0.001)	2.62%

Mean difference between unseen and seen AUC for EDL model is 2.33~2.4%.

## Data Availability

Due to the proprietary nature of this study, supporting data cannot be made openly available.

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
