# Peer review of "Attention-Enabled Ensemble Deep Learning Models and Their Validation for Depression Detection: A Domain Adoption Paradigm"

_diagnostics, 2023, doi:10.3390/diagnostics13122092_

Round 1

Reviewer 1 Report

This was an interesting paper to read. It is well-represented with sufficient reasons for claims. However, there are some minor things that I missed in the paper. Since you are working with deep learning and it is a black box model, I would like to see the results of explainable AI and more general responsible AI in your paper. Furthermore, I noticed that you are using BERT, which is computationally expensive, and its optimized version, liteBERT. What was the reason behind choosing these models? We know that there are many others available

Reviewer 2 Report

Excellent paper!  I have a few minor suggestions:   1. In testing cross-domains, the study shows there is a comprehensive data analysis. Some of the data is obtained from more obscure datasets. For example, Reddit is an anonymous platform so it is unclear how to validate across domains. This could be explained in terms of whether data is authenticated in a process, or not.   2. "To encourage further research in the field and improve current benchmark models, high-quality datasets of substantial size are necessary to build more robust and optimal models". I understand that platforms such as Reddit have been used previously for data mining for mental health. I see social media datasets such as Twitter were used ... is it possible to data mine other more popular social media platforms with video sharing utility? If not, what is holding the study back from looking at more popular platforms such as Facebook, TikTok and YouTube? If so, have these been explored to improve the size and quality of datasets? If there are privacy and security issues, then explain.    3. I suggest showing how research into AI sentiment could have real-world impact in detection followed by moderation. Although the current study is testing accuracy of detection, how can it be useful for informing automated moderation? This could be best described in layman's terms, in future research if applicable.
